# New Therapeutic Approaches in Treatment of Dyslipidaemia—A Narrative Review

**DOI:** 10.3390/ph15070839

**Published:** 2022-07-07

**Authors:** Iveta Merćep, Dominik Strikić, Ana Marija Slišković, Željko Reiner

**Affiliations:** 1Division of Clinical Pharmacology, Department of Internal Medicine, University of Zagreb School of Medicine, 10000 Zagreb, Croatia; 2Division of Clinical Pharmacology, Department of Internal Medicine, University Hospital Centre Zagreb, 10000 Zagreb, Croatia; strikic.dominik@gmail.com; 3Department of Cardiovascular Diseases, University Hospital Centre Zagreb, 10000 Zagreb, Croatia; sliskovic_anamarija@yahoo.com; 4Division of Metabolic Diseases, Department of Internal Medicine, University Hospital Centre Zagreb, 10000 Zagreb, Croatia; zeljko.reiner@kbc-zagreb.hr

**Keywords:** dyslipidaemia, bempedoic acid, PCSK9 inhibitors, pelacarsen, ANGPTL3 inhibitors

## Abstract

Dyslipidaemia is a well-known risk factor for the development of cardiovascular disease, a leading cause of morbidity and mortality in developed countries. As a consequence, the medical community has been dealing with this problem for decades, and traditional statin therapy remains the cornerstone therapeutic approach. However, clinical trials have observed remarkable results for a few agents effective in the treatment of elevated serum lipid levels. Ezetimibe showed good but limited results when used in combination with statins. Bempedoic acid has been thoroughly studied in multiple clinical trials, with a reduction in LDL cholesterol by approximately 15%. The first approved monoclonal antibodies for the treatment of dyslipidaemia, PCSK9 inhibitors, are currently used as second-line treatment for patients with unregulated lipid levels on statin or statin combination therapy. A new siRNA molecule, inclisiran, demonstrates great potential, particularly concerning compliance, as it is administered twice yearly and pelacarsen, an antisense oligonucleotide that targets lipoprotein(a) and lowers its levels. Volanesorsen is the first drug that was designed to target chylomicrons and lower triglyceride levels, and olezarsen, the next in-line chylomicron lowering agent, is currently being researched. The newest possibilities for the treatment of dyslipidaemia are ANGPTL3 inhibitors with evinacumab, already approved by the FDA, and EMA for the treatment of familial hypercholesterolemia. This article provides a short summary of new agents currently used or being developed for lipid lowering treatment.

## 1. Introduction

Cardiovascular disease (CVD) is a leading cause of morbidity and mortality worldwide. Apart from well-known risk factors such as hypertension, diabetes mellitus, and smoking, dyslipidaemia plays an important role in the development of cardiovascular disease; people with elevated serum lipoprotein levels have a two times greater chance of developing CVD than those without [1]. Dyslipidaemias, a crucial correctable predictive factor for coronary artery disease, are defined as alternations of the plasma lipid profile: increased levels of total cholesterol, LDL cholesterol or triglycerides, or a low plasma concentration of HDL cholesterol, or a combination of these features [2]. The global burden of dyslipidemia has increased over the past 30 years, with elevated plasma LDL cholesterol levels being the eighth most important risk factor for death in 2019 [2]. Hypercholesterolemia, the most common form of dyslipidemia, is associated with elevated LDL cholesterol levels, which have been shown to be an important causal factor for ischemic heart disease and ischemic stroke in both developed and developing countries [2]. In 2019, ~4.40 million (95% CI 3.30–5.65 million) deaths and ∼98.62 million (95% CI 80.34–118.98 million) disability-adjusted life years (DALYs; a measure of the overall burden of disease, representing the number of years lost due to disease, disability, or early death) were attributable to high plasma LDL cholesterol levels, and these rates were higher in men than in women [3]. Lifestyle changes, improvements in dietary habits and increased use of lipid-lowering drugs, in part because of their lower cost as most statins are available in generic forms, have led to a decrease in the prevalence of hypercholesterolemia and consequently, a reduction in mean plasma cholesterol levels in high-income countries (especially in Europe and North America) [4,5]. On the other hand, dyslipidaemias are still unrecognized and undertreated in low-income countries where statins are inaccessible to broad segments of the world’s population. From 1980 to 2018, substantial decreases in plasma total cholesterol and non-HDL cholesterol, total cholesterol excluding HDL cholesterol levels have been observed in Central Europe, in both men and women [6]. Additionally, age-standardized death rates (ASDRs) for ischemic heart disease attributable to high plasma LDL cholesterol levels presented per 100,000 of the general population indicated a significant decrease in Central Europe from 1990 to 2019, from 143.38 (181.55–107.26) in 1990 to 71.86 (95.36–49.06) in 2019 for both sexes [3]. This positive trend was also observed in ASDR for ischemic stroke attributable to high plasma LDL cholesterol levels from 24.81 (51.27–8.54) in 1990 to 14.14 (30.46–4.23) in 2019 [3]. Among regions in Central Europe, the highest ASDR for IHD attributable to high plasma LDL cholesterol levels was found in Bulgaria (108.81), and the lowest in Slovenia (27.53) [3]. A concerning fact, approximately 50% of American adults have elevated LDL cholesterol levels, and it is estimated that less than 35% of these patients achieve the target LDL levels as recommended by the guidelines [7]. 

Dyslipidaemias are divided into primary or familial dyslipidaemias, or secondary to other conditions (such as diabetes mellitus, thyroid diseases, obesity, or an unhealthy lifestyle), the latter being more common [2]. Familial forms account for less than 2% of all dyslipidaemias. Primary dyslipidaemias are caused by a genetic defect of a single gene (monogenic) or multiple genes (polygenic) [8]. Almost 50 years ago, Donald S. Fredrickson and his colleagues recognized alternations of physiologic levels of plasma lipoproteins that paved the way for the first phenotypic classification of lipoprotein disorders, established by separating lipoproteins using density gradient ultracentrifugation (beta-quantification). According to this classification, dyslipidaemias are subdivided into five different categories. In type I, primary hyperlipoproteinemia or familial hyperchylomicronaemia, caused by lipoprotein lipase deficiency or altered Apo-CII, there is an abnormality of chylomicrons leading to triglycerides greater than 99 percentiles; it is very rare. In type IIa, familial hypercholesterolemia or polygenic hypercholesterolemia, there is an LDL receptor deficiency resulting in elevated LDL cholesterol together with a total cholesterol concentration greater than 90 percentile and often apolipoprotein B, greater than 90 percentile. The most common phenotype IIb, familial combined hyperlipidaemia, consists of an abnormality in LDL and very-low-density lipoprotein (VLDL) cholesterol, affecting total cholesterol and/or triglycerides more than 90 percentile, and apolipoprotein more than 90 percentile. Type III, familial dysbetalipoprotenaemia, caused by a defect in Apo E 2 synthesis, is a rare form in which there is an abnormality in VLDL remnants and chylomicrons, resulting in elevated IDL; total cholesterol and triglycerides are greater than 90 percentile. Familial hypertriglyceridemia, phenotype IV, is caused by increased VLDL production and decreased excretion, leading to total cholesterol and triglycerides greater than 90 percentile. Finally, type V or endogenous hypertriglyceridemia is characterized by abnormal chylomicrons and VLDL, and triglycerides are greater than 99 percentiles [8,9,10]. It is important to mention that the Fredrickson classification system does not address dyslipidaemias related to low HDL cholesterol or elevated Lp(a). For this and other reasons, this classification is no longer so commonly used, and dyslipidaemias are usually defined as those with elevated total and LDL cholesterol, those with elevated triglycerides, and those with both elevated LDL cholesterol and elevated triglycerides, with considerable attention also paid to elevated Lp(a) and low HDL cholesterol. In evaluation of primary dyslipidaemias, family history is of great importance, together with a physical examination which can often reveal xanthomas, deposits of lipids in the skin and sometimes subcutaneous tissue. Currently, secondary dyslipidaemias related to diabetes mellitus, obesity, an unhealthy lifestyle, and other medical disorders are the focus of interest because dietary and lifestyle modifications influence these conditions to a great extent.

Lipid-lowering therapy, primarily statins, also known as hydroxymethyl glutaryl coenzyme A reductase inhibitors, decrease vascular events including stroke and myocardial infarction and cause a reduction in all-cause mortality by 10% for every 1.0 mmol/L reduction in LDL-cholesterol levels [11]. Therefore, statins are well-established and widely recommended therapeutics, according to the guidelines of the European Society of Cardiology and the American Heart Association. Apart from statins, other therapeutics that reduce LDL levels have been shown to be effective in decreasing the risk of cardiovascular events. Ezetimibe binds to the NPC1L1 transporter protein and consequently inhibits intestinal absorption of cholesterol, and has a synergistic effect when combined with statins in reducing LDL cholesterol levels and thereby MACE (major cardiovascular events), as shown in the IMPROVE-IT and SHARP trials [12,13]. Adding proprotein convertase subtilisin/kexin type 9 (PCSK9) inhibitors to moderate- or high-intensity statin therapy regimens leads to cardiovascular risk reduction in patients with stable atherosclerotic cardiovascular disease, or after recent acute coronary syndromes [14,15]. These medicines inactivate proprotein convertase subtilsin/kexin type 9 (PCSK9), inhibiting the labelling of LDL receptors for degradation, thus prolonging LDL receptor activity at the cell membrane, leading to additional LDL cholesterol, lowering by 50% to 60% compared to statin monotherapy [16]. 

In 2018, the American College of Cardiology/American Heart Association published lipid-lowering guidelines that emphasize patient-tailored therapy and personalized risk stratification [17]. Since then, new lipid-lowering therapeutics have been evaluated. Among the newly developed drugs is inclisiran, a small interfering RNA targeting hepatic PCSK9 synthesis led to increased LDL receptor recycling and potent LDL cholesterol reduction by 51% when used together with other lipid-lowering agents [18,19]. Another potentially promising molecule is evinacumab, a monoclonal antibody that inhibits ANGPTL3, causing LDL cholesterol reduction regardless of LDL receptor function [20]. Furthermore, studies evaluating vupanorsen, a drug that targets ANGPTL3 with an antisense oligonucleotide, showed statistically significant dose-dependent reductions in LDL cholesterol, triglycerides, non-HDL cholesterol, and total cholesterol [21]. Other therapeutic possibilities include ARO-ANG3, siRNA directed against ANGPTL3 mRNA currently in the phase II clinical trial and bempedoic acid, which inhibits ATP-citrate lyase, both of which successfully reduce LDL cholesterol levels [18,22,23]. 

The aim of this article was to review new therapeutic approaches for the treatment of dyslipidaemia with medication options that have passed through phase II clinical trials. 

## 2. Methods

This narrative review was performed by searching PubMed, Scopus, Embase, and Web of Science from the inception to April 10, 2022 using the following key words: dyslipidaemia, lipid lowering drugs, treatment strategies, statins, PCSK9 inhibitors, bempedoic acid, ezetimibe, inclisiran, pelacersen, volanesorsen, olezarsen, mipomersen, lomitapide, lerodalcibep, ANGPTL3 inhibitors.

## 3. Results

### 3.1. Drugs Used in Dyslipidaemia

#### 3.1.1. Statins

Statins became almost a synonym for the treatment of dyslipidaemia. These drugs are among the most widely used medications, with atorvastatin being one of the best-selling pharmaceuticals [24]. Statins have been in use for almost 50 years and are the first therapeutic choice for dyslipidaemia, particularly for elevated LDL cholesterol levels [25]. Current statins approved by the EMA include atorvastatin, simvastatin, rosuvastatin, fluvastatin, pitavastatin, lovastatin and pravastatin. They act as antagonists for 3-hydroxy-3-methylglutaryl coenzyme A reductase, and by inhibiting the enzyme HMG-3-CoA, reductase blocks the endogenic cholesterol synthase pathway, resulting in lower LDL cholesterol serum levels [26]. The main indication for statin treatment is a high serum cholesterol level and CVD. Studies have observed beneficial effects of statins on atherosclerotic plaques by stabilizing plaques and decreasing the risk of cardiovascular incidents [27]. Statins are generally well-tolerated drugs, but they have significant adverse effects, including myopathy and rhabdomyolysis. In addition, there is no doubt that long-lasting statin treatment results in de novo occurrence of type 2 diabetes mellitus (T2DM), particularly in subjects who have metabolic syndrome or are prone to T2DM. However, recent studies provide insight to these observations with potential benefits of statin therapy by far exceeding the risk of new onset T2DM. Claims of connection of long-term statin treatment with new neoplastic processes were disproved and claims of cognitive deterioration while on statin treatment could not be proven in recent studies [28]. Statins remain the first choice of medication for the treatment of dyslipidaemia. However, a new generation of lipid-lowering drugs has been introduced annually with statin treatment expanding from monotherapeutic to politherapeutic, in combination with other lipid-lowering drugs. 

#### 3.1.2. Bempedoic Acid

Bempedoic acid is an adenosine triphosphate (ATP) citrate lyase inhibitor that inhibits cholesterol biosynthesis and increases LDL receptor expression. ATP citrate lyase is an enzyme upstream of HMG-CoA reductase in the biochemical cholesterol synthesis pathway. Inhibition of ATP citrate lyase prevents endogenous cholesterol synthesis and indirectly increases the expression of LDL receptors, thereby increasing the clearance of LDL cholesterol [29]. Bempedoic acid is a prodrug that requires activation, and the active metabolite ESP15228 inhibits ATP citrate lyase [30]. It is administered orally once a day at a dose of 180 mg and is approved by the FDA and EMA for the treatment of hypercholesterolemia [30,31]. Efficacy and safety of bempedoic acid have been evaluated in five clinical trials: HARMONY, WISDOM, SERENITY, TRANQUILITY and OUTCOMES. Currently, the OUTCOMES study is underway with a focus on the reduction in cardiovascular incidents and diabetes compared to patients on statin therapy, a thesis that is supported by literature reviews published so far [32]. The other four studies showed the beneficial effects of lowering LDL cholesterol levels. In the HARMONY and WISDOM studies, patients on oral bempedoic acid were evaluated with a control group of patients on high-dose statin therapy. Results showed 16.5% and 15.1% reductions in LDL cholesterol levels in the blood respectively, after 12 weeks of therapy [33,34]. The TRANQUILITY study provided a significant safety profile for bempedoic acid, and it is well-tolerated. An important finding was the lack of myopathies that are sometimes observed with prolonged statin therapy [35]. Most common side effects include upper respiratory tract infection, urinary tract infection, arthralgia, muscle spasms and diarrhoea; however, it is worth mentioning that similar occurrences of these symptoms were observed in the placebo groups [30]. In the SERENITY trial, one observation was an increase in uric acid levels in the blood, but other trials did not support this finding [23]. 

#### 3.1.3. Ezetimibe

Ezetimibe is a selective inhibitor of cholesterol absorption. The mechanism of action of ezetimibe is mediated by targeting the sterol transporter Neimann-Pick C1 Like 1 (NPC1L1), which is localized at the border cells in the small intestine. Binding to the transporter inhibits it and decreases the absorption of cholesterol, further decreasing cholesterol circulation through the liver, and finally increasing the clearance of cholesterol from blood [30]. It is orally administered at a daily dose of 10 mg. It is worth noting that the absorption of ezetimibe differs in younger and older adults, with older adults having greater exposure than younger adults. However, no dose correction was required in older adults [36]. The efficacy of ezetimibe is mostly observed in co-therapy with statins, with LDL cholesterol levels decreasing by 10 to 15%, varying with the different statin agents combined. More recent studies have shown great results in ezetimibe and bempedoic acid co-therapy, with a mean difference of 38% in LDL cholesterol levels compared to the placebo group. Ezetimibe monotherapy is also acceptable, especially in patients who do not tolerate statins and require modest LDL cholesterol level reduction, with results showing an 18% decrease compared to placebo patients [31,37]. The FDA and EMA approved ezetimibe and currently, there are fixed combinations of different statins with ezetimibe and fixed combinations of bempedoic acid 180 mg with ezetimibe 10 mg. Approval is for the treatment of hypercholesterolemia and mixed dyslipidaemia in patients at a high risk of cardiovascular incidents with high LDL cholesterol levels. Ezetimibe has a good safety profile, with few to no adverse effects noted in the literature. The SEAS trial observed a potentially increased incidence of cancer in patients receiving ezetimibe therapy, but further trials found no significant difference [30,38].

#### 3.1.4. PCSK9 Inhibitors

Proprotein convertase subtilisin/kexin type 9 (PCSK9) inhibitors are a new generation of lipid-lowering drugs with many clinical trials suggesting very good LDL cholesterol-lowering results. PCSK9 plays an important role in LDL receptor downregulation. LDL receptors are found on hepatocytes and play a role in the removal of circulating LDL cholesterol from blood. When the PCSK9 protein binds to the LDL receptor, it starts the process of degrading the receptor, thus increasing LDL cholesterol levels. The monoclonal antibodies alirocumab and evolocumab inhibit PCSK9 binding to LDL receptors, increase recycling of LDL receptors, and indirectly lower circulating LDL cholesterol levels by increasing LDL cholesterol uptake [39]. The third monoclonal antibody in this drug class was bococizumab, but trials were discontinued because of immunogenicity and higher injection site reactions than in two aforementioned [30]. Both currently used monoclonal antibodies, alirocumab and evolocumab, are administered subcutaneously once every two weeks in doses starting from 75 mg to 300 mg depending on the initial LDL cholesterol levels and 140 mg, respectively. It is important to note that steady concentrations of alirocumab and evolocumab were achieved within 4–6 weeks from the start of the treatment [40,41]. Most insights on PCSK9 inhibitors were obtained in the ODYSSEY OUTCOMES trial of alirocumab and the FOURIER trial of evolocumab. The ODYSSEY OUTCOMES trial showed that alirocumab not only lowers circulating LDL cholesterol levels but also has a preventive effect on cardiovascular incidents [41]. The FOURIER trial observed a 15% decrease in cardiovascular incidents compared with the placebo group in patients receiving evolocumab therapy, who previously had major cardiovascular incidents and were on high-dose statin therapy [40]. The above-mentioned trials provided important information on the PCSK9 inhibitor safety profile. Both monoclonal antibodies are safe, with the most common adverse effect being an injection site reaction. In addition, no cognitive performance alteration or new-onset diabetes was observed with the use of alirocumab or evolocumab [42]. The FDA and EMA approved both monoclonal antibodies for use in patients with high cardiovascular incident risk and high LDL cholesterol levels. They are mostly used in combined therapy with statins, or with statins plus ezetimibe, but also as a monotherapy in patients who are intolerant or resistant to statin therapy. The most limiting factor for the wider use of PCSK9 inhibitors at this moment is their cost. However, even today the benefits of these monoclonal antibodies surpass their relative high cost for certain indications [30,42].

#### 3.1.5. Inclisiran

One of the most recently approved drugs for the treatment of dyslipidaemia is inclisiran. It is a small interfering ribonucleic acid (siRNA) that targets PCSK9. As mentioned earlier, PCSK9 is an important protein involved in LDL receptor degradation. Inclisiran, an siRNA, interferes with the translation of PCSK9 by cleaving messenger RNA, thereby decreasing PCSK9 production. The absence of PCSK9 results in the upregulation of LDL receptors and, consequently, lowers the circulating level of LDL cholesterol [19,42]. The biggest advantage of inclisiran is its administration scheme. Inclisiran is administered subcutaneously on days 1, day 90, day 180, and afterwards once every six months. The usual dosage is 284 mg in a single administration [19]. Efficacy of inclisiran was observed in ORION trials. These trials showed that LDL cholesterol levels were reduced by approximately 50% compared to placebo groups [31]. Although these results indicate slightly lower efficacy than that of monoclonal PCSK9 inhibitors, patient compliance with inclisiran therapy is believed to be better because of the infrequent administration scheme [19]. In addition, the ORION studies provided information about the safety profile of inclisiran. It is well tolerated and safe for use in all patients, without dosage correction. Adverse effects are uncommon with inclisiran therapy, but we must acknowledge that local reactions at the administration site, muscular pain and upper respiratory tract symptoms are common. However, ORION studies observed the same frequency of aforementioned adverse effects in the inclisiran and placebo groups [19,30]. Inclisiran has been approved by the FDA and EMA for the treatment of mixed dyslipidaemia and hypertriglyceridemia.

#### 3.1.6. Pelacarsen 

The discovery of lipoprotein(a) [(Lp(a)], and the fact that elevated Lp(a) is an independent risk factor for atherosclerosis and CVD, directed the research to find drugs which might decrease its levels in serum and therefore could prevent cardiovascular events in patients with elevated Lp(a) levels. Lp(a) is in fact an LDL cholesterol variant that contains an apolipoprotein(a) [apo(a)] [43]. Pelacarsen was one of the first drugs in class to be evaluated in humans and is currently in the most advanced phase of clinical trials [44]. It is an antisense oligonucleotide that binds to hepatocyte apo(a) mRNA and forms an ASO/mRNA complex that prevents the translation of apolipoprotein(a). This leads to decreased apolipoprotein(a) production and, by default, lower circulating Lp(a) levels [45]. Clinical trials of pelacarsen are still underway, but results published so far are rather optimistic, with phase II clinical trials already completed, and phase III currently in the recruitment stage. Phase II observed dose-dependent and regiment-dependent reductions in Lp(a) levels compared to the placebo group. Pelacarsen was administered subcutaneously at a dosage scheme of 20 mg every 4 weeks, 20 mg every 2 weeks, 20 mg every week, 40 mg every 4 weeks, and 60 mg every 4 weeks, with an observed reduction in Lp(a) circulating levels by 35%, 58%, 80%, 56% and 72%, respectively. Most adverse effects observed were mild to moderate, with 5% of patients discontinuing pelacarsen treatment because of myalgia, arthralgia, malaise, or injection site reactions. Two deaths occurred in the group receiving pelacarsen treatment, both in unforeseen circumstances: one due to a road traffic accident and the second as a result of suicide in the patient with underlying depression [46]. Most recent studies have observed lower levels of not only Lp(a), but also lower levels of oxidized phospholipids on apolipoprotein(a) and apolipoprotein(b), lower levels of apolipoprotein(b), and LDL cholesterol. However, the true meaning of this result is questionable as current measurement of LDL cholesterol considers Lp(a) levels, so lower levels of LDL cholesterol might be due to a relative reduction on a count of absolute reduction in Lp(a) [44]. Pelacarsen is still not an approved drug with phase III clinical trials pending, but results of the trials published so far indicate that Lp(a) lowering certainly led to a decrease in cardiovascular incidents. The phase I trial of another siRNA molecule olpasiran showed a reduction in Lp(a) for over 90% and very few adverse effects compared with a reduction of up to 80% with pelacarsen in a phase II trial [46,47]. In a small phase I trial of the siRNA SLN360, which enrolled 32 participants with elevated Lp(a) levels and no known CVD, the maximal median percentage change from baseline in Lp(a) level over 150 days was observed: −10%, −46%, −86%, −96%, and −98% for the placebo group and the 30-mg, 100-mg, 300-mg, and 600-mg SLN360 groups, respectively [48].

#### 3.1.7. Volanesorsen

Management of dyslipidaemia prior to the present day has mostly focused on LDL cholesterol. Currently, the focus is being switched to other particles such as triglycerides and triglycerides rich particles such as chylomicrons [49]. Chylomicrons are ultra-low-density lipoproteins that mostly contain triglycerides. An important protein found in chylomicrons is the apolipoprotein C-III. Numerous studies have demonstrated the importance of apoC-III in increasing circulatory triglyceride levels and intrinsic proatherogenic effects [45]. As such, apoC-III is a target for a new class of drugs aimed to lower triglyceride levels and indirectly preventing cardiovascular incidents and pancreatitis episodes in patients with hypertriglyceridemia. The first of these drugs is volanesorsen. It is an antisense oligonucleotide that binds to the ApoC-III mRNA and disrupts apoC-III translation. This leads to lower apoC-III levels and lower levels of chylomicrons and triglycerides [50]. Phase III clinical trial results for volanesorsen were published in 2021, with an observed reduction in triglyceride levels by 71.8% compared to the placebo group after a three-month period. In addition, pancreatitis events were reduced, with five acute pancreatitis events in the placebo group versus none in the volanesorsen treatment group. It is worth mentioning that patients eligible for participation in the Phase III clinical trials were those diagnosed with Familiar Chylomicronaemia Syndrome (FCS) [51]. Based on these results, volanesorsen was approved by the EMA for the treatment of hypertriglyceridemia and FCS. Even though the results of volanesorsen are rather optimistic, the safety profile has yet to be completely clarified. The most common adverse effect of volanesorsen is the injection site reaction; however, serious thrombocytopenia has been observed. The effect of thrombocytopenia is yet to be determined, but patients on volanesorsen treatment should be closely monitored for a potential reduction in platelet count [50,51]. Currently, volanesorsen is administered subcutaneously at a dose of 285 mg once a week. Re-evaluation of therapy should be considered after three months and patients with a good response to treatment may reduce the dosage scheme to once every two weeks. The FDA approval of volanesorsen is pending at this moment. 

#### 3.1.8. Olezarsen

Similar to volanesorsen, olezarsen is an antisense oligonucleotide targeting apoC-III, and through apoC-III translation disruption, chylomicrons and triglyceride levels are reduced. It is currently a next-generation antisense medicine in phase III clinical trials. The first results of the Phase II trial were published in early 2022. The treatment regimen tested was 10 mg every 4 weeks, 15 mg every 2 weeks, 10 mg every week, and 50 mg every 4 weeks. Olezarsen is administered subcutaneously. Evaluation was performed after 6 months of treatment with a reduction in triglycerides in patients receiving olezarsen treatment by 23%, 56%, 60%, and 60%, respectively. The safety profile seems to be the biggest benefit of olezarsen compared to volanesorsen, with no platelet count reduction and only mild injection site reactions as the main adverse effects observed [52].

#### 3.1.9. Other Agents Approved by FDA

One of the first drugs developed for parenteral use in dyslipidaemia was mipomersen, an antisense oligonucleotide that targets apolipoprotein B (Apo B) mRNA and interferes with translation, thereby decreasing Apo B levels [53]. Apolipoprotein B is one of the main components of LDL and VLDL particles. The absence of Apo B decreases LDL cholesterol and total cholesterol levels in the blood. In studies, a 24.7% reduction in LDL cholesterol levels was observed over 26 weeks in patients receiving mipomersen compared to patients receiving a placebo. In 2012, the FDA approved mipomersen for subcutaneous administration at a dose of 200 mg once weekly in patients with homozygous familial hypercholesterolemia [54]. However, mipomersen had significant adverse effects ranging from local injection reactions to flu-like symptoms, causing many patients to discontinue treatment. The most serious adverse effect is liver toxicity, with an increase in liver enzymes and accumulation of liver fat [54,55,56]. Therefore, mipomersen was never approved by the EMA after two unsuccessful authorization attempts, and after initial approval by the FDA, mipomersen received a special hepatotoxicity warning and is available only in a restricted program. Another important agent that is still in use is lomitapide, an inhibitor of the microsomal triglyceride transfer protein (MTP). This protein is required for the transfer of triglycerides to Apo B and the assembly of VLDL particles. Inhibition of MTP causes a reduction in Apo B secretion in the liver and a reduction in total cholesterol levels [55]. Lomitapide showed a significant reduction in LDL cholesterol, both as monotherapy and as a combination therapy with ezetimibe. It is administered orally at a dose of 5–10 mg [57]. Lomitapide is generally safe to use, with the most serious adverse effect being an increase in liver enzymes [58]. Both the FDA and EMA have approved lomitapide for the treatment of homozygous familial hypercholesterolemia.

#### 3.1.10. Treatments in the Pipeline

Another substance currently in development targeting apo C-III, is an apo C-III short interfering RNA antagonist with the acronym ARO-APOC3 [59]. The preliminary results of a phase I trial evaluating the safety and key pharmacodynamics and lipid parameters in patients with severe hypertriglyceridemia, ARO-APOC3 caused a maximal mean reduction of 80% to 99% in apo C-III levels, and a maximal mean reduction of 74% to 92% in plasma triglyceride levels. Since an E3-ubiquitin ligase, an inducible degrader of LDL receptor (IDOL) can cause ubiquitination and degradation of LDL receptors in lysosome, potentially a novel regulator of LDL receptor expression, similar to PCSK9. Although there are so far no approved drugs for targeting the IDOL-LDL-receptor pathway, some recent studies have shown that IDOL might be a potential therapeutic target for treatment of hypercholesterolemia [60].

#### 3.1.11. ANGPTL3 Inhibitors

One of the new possible targets for the treatment of dyslipidaemia is angiopoietin-like 3 protein (ANGPTL3), which is currently one of the main focal points of lipidology pharmaceutical studies. So far, different teams have attempted to target ANGPTL3 using different technologies such as monoclonal antibodies and antisense oligonucleotides. ANGPTL3 acts as an inhibitor of lipoprotein lipase (LPL) and endothelial lipase (EL) enzymes [61]. Both enzymes are important in the serum increase in triglycerides and LDL cholesterol, so inhibition of ANGPTL3 protein leads to the disinhibition of LPL and EL, and therefore a decrease in triglycerides and LDL cholesterol levels in circulation [62,63]. The first drug in the class of ANGPTL3 inhibitors is evinacumab, a monoclonal antibody already approved by the FDA and EMA for the treatment of familial hypercholesterolemia. It is administered intravenously every four weeks at a dose of 15 mg/kg body weight. Efficacy of evinacumab was observed in a Phase III clinical trial with a decrease in 47,1% in serum LDL cholesterol levels in patients receiving evinacumab therapy, compared to a 1,9% increase in placebo group patients. Evinacumab has very few reported adverse effects, with most being upper respiratory tract infections and flu-like syndromes. Like other intravenously administered drugs, evinacumab has the potential to cause serious allergic reactions or even anaphylaxis [20]. Another possibility to target ANGPTL3 is the antisense oligonucleotide vupanorsen. Vupanorsen is a liver-targeted antisense oligonucleotide that inhibits the synthesis of ANGPTL3. Initially, vupanorsen showed promising results in decreasing serum LDL cholesterol levels. However, the clinical trial TRANSLATE-TIMI 70 was discontinued in early 2022 because of serious hepatic steatosis, and an increase in alanine aminotransferase (ALT) and aspartate aminotransferase (AST) were observed as adverse effects [64]. 

#### 3.1.12. Lerodalcibep

Lerodalcibep inhibits PCSK9 by gene editing, using CRISPR-Cas9 techniques. It is a recombinant fusion protein of a PCSK9-binding domain (adnectin) and human serum albumin that increases its half-life to 12–15 days. This enables the administration in a small-volume injection only once a month [65]. A phase II study with this drug at a dose of 300 mg once a month in patients who had LDL cholesterol >2.0 mmol/L (~80 mg/dL) despite maximally tolerated statin treatment, decreased LDL cholesterol by more than 70% at the end of the 12-week study. The incidence of adverse events was almost equal to the one in the placebo group. An extension of this study showed a stable mean 60% reduction in LDL cholesterol over 36 weeks. The ORE trial program focusing on homozygous familial hypercholesterolemia, proven CVD, and high-risk primary prevention is ongoing and consists of six phase III trials [65]. 

#### 3.1.13. Vaccines against PCSK9 

Another attractive strategy is based on vaccines against PCSK9, which should trigger the generation of host anti-PCSK9 antibodies and consequently neutralize PCSK9/LDL receptor interactions. A novel antiPCSK9 vaccine formulation, called liposomal immunogenic-fused PCSK9-tetanus peptide plus alum adjuvant (L-IFPTA), was recently designed. The efficacy of the L-IFPTA vaccine, containing two different epitopes belonging to PCSK9 and tetanus toxin proteins, has been observed in different animal models including mice and non-human primates [66]. For instance, it induced the production of long-lasting functional and safe anti-PCSK9 specific antibodies in BALB/c mice and reduced LDL cholesterol and VLDL cholesterol levels by up to 51.7 and 19.2% in C57BL/6 mice, without any significant adverse events [67].

## 4. Discussion

According to the latest guidelines for the treatment of dyslipidaemia published by the European Society of Cardiology, statins remain the first therapeutic option in the treatment of dyslipidaemia, with study results demonstrating up to a 23% reduction in major coronary events and a 10% reduction in all-cause mortality over 5 years [68]. No such results have been observed with any of the other agents mentioned above in terms of reducing major coronary and cardiovascular events, this being due to the lack of studies using only the agent in question, without statin treatment, and the lack of studies focusing on cardiovascular prevention rather than just cholesterol lowering. Given the widespread use of statins and the fact that LDL levels are trending downwards, with current guidelines recommending LDL levels of <1.4 mmol/L for both primary and secondary prevention in high-risk patients, the “lower is better” strategy raises the question of what the target LDL levels should be in these patients in the future, when there is currently no evidence that LDL < 1 mmol/L is harmful. Although statins rarely cause serious side effects including muscle damage, there is still public concern that statins may cause less severe muscle symptoms. However, placebo-controlled randomised trials have shown that true statin intolerance is rare, so other statins can be used or the dose reduced in the majority of patients at high risk of major cardiovascular events [28]. Weighing up the risks and benefits, discontinuation of therapy in these patients would lead to serious consequences, the treatment of which would further burden the healthcare system. Nevertheless, more data are needed from randomized controlled trials and observational studies on the cost-effectiveness of cardiovascular disease prevention through lipid modification. In addition, risk assessment needs to be better aligned in the future, as the estimation of total cardiovascular risk with the SCORE system is based on total cholesterol, whereas screening, recommendations and treatment are based on LDL [68]. Thanks to advances in treatment, the proportion of elderly patients is increasing, especially those aged > 80 years. Trials should therefore be encouraged to support the use of statins in this population, especially in primary prevention. Ezetimibe is the second-line treatment in combination with statins, with PCSK9 inhibitors being the next choice when combined treatment with ezetimibe and statins is insufficient or patients are at high risk for major cardiovascular events [68]. Given the continued clinical benefit of early initiation of statin treatment in patients with acute coronary syndrome or stroke, the optimal timing of treatment with PCSK9 inhibitors should be investigated in dedicated outcome assessment trials.

At this point, it is worth noting that this study is based on limited published data on different lipid-lowering agents, most of which were observed in controlled populations already treated with lipid-lowering agents during clinical trial phases. In addition, this review was searched through online open access databases, so we may have missed some of the previously published studies. Finally, the latest guidelines are from 2019, and since then, new agents have been approved and used in lipid-lowering treatment, so we can expect changes to the guidelines in the future.

## 5. Conclusions

The pharmacology of dyslipidaemia is one of the fastest-growing fields in pharmaceutical development. Currently, there are numerous new agents aimed at lowering lipid levels and thus decreasing the morbidity and mortality of cardiovascular disease, one of the leading causes of premature death in patients in the developed world. However, dyslipidaemias remain a global health problem that should foster collaboration among key scientific institutions and health care providers. The medical community has treatment guidelines for dyslipidaemias, but we need to revise them to include country- and region-specific lipid-lowering approaches and cardiovascular disease prevention programs. As new clinical trial results are published on an almost daily basis and regulatory agencies discus new drug approvals, we can only follow publications and evaluate patients with an increased risk of cardiovascular incidents in search of the best treatment option for individual patients’ comorbidities and their lifestyle. Whether some of the mentioned agents will be approved is still unknown, but we can surely look forward to finding new therapeutic approaches for the treatment of dyslipidaemia, most likely a combination of existing drugs and maybe some which are still in the pipelines. It seems that today, particularly when LDL cholesterol lowering is concerned, the words of the last author of this article written more than a decade ago sound quite prophetical: “It could be hypothesized that although today most of the patients with dyslipidaemia are treated with statins as monotherapy, this will change in the future. The pattern might follow what happened with antihypertensive treatment which was also several decades ago based upon monotherapy. However, current combinations of two, three or even four drugs lowering the blood pressure by influencing different blood-pressure-lowering mechanisms are widely used to achieve the treatment goals as defined by the guidelines”. 

## Data Availability

Data sharing not applicable.

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
