# Peer review of "New Therapeutic Approaches in Treatment of Dyslipidaemia—A Narrative Review"

_pharmaceuticals, 2022, doi:10.3390/ph15070839_

Round 1
Reviewer 1 Report
Dear authors,
The manuscript presents an interesting overview of new agents used or developed for the treatment of dyslipidaemia.
Please find my suggestions and comments below:
- Lines 22-23: “This narrative review article provides a short summary of new agents currently used or being developed for the treatment of dyslipidaemia”. According to authors, I consider that this manuscript is a short summary of lipid lowering drugs. Authors should provide many details about each drug presented in the manuscript, in order to be included in the Review category. In my opinion, in this form, this manuscript could be included in Mini review category.
- Authors should include in this manuscript other drugs, such as: mipomersen, lomitapid, etc.
- Line 11: please check if “fortunately” is correctly used.
- Introduction:
o Worldwide statistics regarding the dyslipidemia (no of patients, incidence, mortality, etc.) should be provided.
o It would be better if authors present a short description of physiopathology and types of dyslipidemia.
- Discussion:
o A figure with mechanisms of action should be inserted in the manuscript
o A classification of these drugs according to their mechanism of action should be present. You can use a table. Also, you could present the influence of these agents on lipoproteins level and their indication (type of dyslipidemia).
o I recommend splitting the presentation of drugs in two main sections:
1) Drugs used in dyslipidemia
2) Treatments in pipeline
o What are the clinical differences between statins? What are their advantages and disadvantages compared to other classes? How they could be use in therapeutics? Authors should also present the results of clinical studies regarding the statin benefits.
o Line 69: lovastatin should be mentioned, too.
o Line 105: please revise “of ezetimibe defers in younger and older adults…”
o A summary table with clinical use and doses of each lipid lowering agent should be inserted
o Authors should present the expected clinical benefits of low-density lipoprotein cholesterol-lowering therapies and the intensity of the treatment used. You can use the clinical guidelines.
o Line 257: “being published daily…”. Please check and revise this aspect.
o Authors should discuss the limitations of the study
Author Response
Dear Reviewer,
We are thankful for your comments and we have gladly accepted them.
Please find revised manuscript with track changes
Iveta Mercep
Reviewer 2 Report
The authors present a narrative review on new treatments for dyslipidemia.
Understanding that a narrative review is presented, which implies that there is no predetermined research question, nor a specific search strategy, and that there is no requirement for protocols to guide the review, I suggest that the following sections be complemented.
1. The methodology is extremely simple, it only presents information on how a search was conducted, however, it is important to also describe which were the key words, which were the basic criteria for the selection of the articles consulted.
2. It is important to present a correlational analysis of the treatments consulted, advantages and disadvantages compared to the first-line treatment and/or other treatments consulted
Author Response
Dear Reviewer,
We have read your comments and we have gladly accepted them.
We are here for any further questions.
Iveta Mercep
Round 2
Reviewer 1 Report
The authors have been partially addressed my recommendations or requests. In my opinion, this manuscript could be included in Mini review category
Some aspects must be detailed:
- Authors should explain the term “non-HDL cholesterol”
- Line 68: Please check the free spaces
- Lines 266-267: Reference 57 is not correct cited for the FDA withdrawal of mipomersen
- In the previous revision I make some recommendations, that I did not identify in the new manuscript:
o I recommended to present a classification of lipid lowering drugs and a figure with mechanism of action of each category of drugs.
o What are the clinical differences between statins? What are their advantages and disadvantages compared to other classes? How they could be use in therapeutics? Authors should also present the results of clinical studies regarding the statin benefits.
o A summary table with clinical use and doses of each lipid lowering agent should be inserted
o Authors should present the expected clinical benefits of low-density lipoprotein cholesterol-lowering therapies and the intensity of the treatment used. You can use the clinical guidelines.
o Authors should discuss the limitations of the study
Reviewer 2 Report
The authors have changed the focus of the manuscript from "narrative review" to "An overview", which implies some important changes, especially in the Methodology section.
The authors should describe precisely the methodology, how many manuscripts were consulted, what were the inclusion criteria and what were the exclusion criteria. How many articles were consulted in each of the databases. I recommend that authors review the guidelines for the development of a review.
The narrative review did not require the development of many of these aspects, due to the purpose of this type of review, but a review is a much more complex structure that requires a detailed presentation of the procedures.
